# Changes in Blood Pressure Variability in Children with Postural Tachycardia Syndrome

**DOI:** 10.3390/children10071244

**Published:** 2023-07-19

**Authors:** Shuangshuang Gu, Shuo Wang, Yuwen Wang, Juan Zhang, Hong Cai, Runmei Zou, Cheng Wang

**Affiliations:** 1Department of Pediatric Cardiovasology, Children’s Medical Center, The Second Xiangya Hospital, Central South University, Changsha 410011, China; vespergu@icloud.com (S.G.); wangyw@csu.edu.cn (Y.W.); zhangjuan001@csu.edu.cn (J.Z.); ralise@csu.edu.cn (H.C.); zourm0721@csu.edu.cn (R.Z.); 2Department of Pediatrics, Xiangya Hospital, Central South University, Changsha 410008, China; wangshuo998@sina.com

**Keywords:** 24 h ambulatory blood pressure monitoring, blood pressure, blood pressure variability, head-up tilt test, children

## Abstract

(1) Objective: In this research, we explored the difference in blood pressure variability (BPV) between children with postural tachycardia syndrome (POTS) and healthy children. Furthermore, we tried to investigate the effect of BPV on POTS and its relationship with prognosis of POTS. (2) Methods: 47 children with POTS (11.2 ± 1.8 years, 23 males) were enrolled in the POTS group and 30 healthy children (10.9 ± 1.9 years, 15 males) were matched for the control group. All participants completed 24 h ambulatory blood pressure monitoring (24hABPM). Thirty-three children with POTS were followed up for 52.0 (30.5, 90.5) days and were divided into a response group and a non-response group after evaluation. (3) Results: The 24 h diastolic blood pressure standard deviation (24hDSD), daytime diastolic blood pressure standard deviation (DDSD), nighttime systolic blood pressure standard deviation (NSSD), daytime diastolic blood pressure variation coefficient (DDCV) and nighttime systolic blood pressure variation coefficient (NSCV) in the control group were lower than those in the POTS group (*p* < 0.05). Percentage of females, age and height were lower in the response group than in the non-response group in children with POTS (*p* < 0.05). Univariate analysis showed that 24hDSD, DDSD, NSSD, DDCV and NSCV were potential risk factors for POTS, and sex and height were potential risk factors for poor prognosis of POTS. After adjusting for covariates, the risk of POTS increased by 48%, 53% and 49% when DDSD, NSSD and NSCV increased by 1 mmHg, 1 mmHg and 1%, respectively. The risk of poor prognosis in females was 12.99 times higher than that in males. (4) Conclusions: The results suggest that children with POTS may have an abnormal circadian rhythm in blood pressure and increased BPV. DDSD, NSSD and NSCV are independent risk factors for POTS, and being female is an independent risk factor for poor prognosis of POTS.

## 1. Introduction

Postural tachycardia syndrome (POTS) is a common hemodynamic type of orthostatic intolerance (OI) in children and adolescents [1]. POTS is a chronic OI with a prevalence of about 0.2%, mostly in people between 15 and 25 years of age, more than three quarters of whom are female [2]. The signs and symptoms are tachycardia without hypotension, along with light-headedness, syncope, palpitations, blurred vision, tremulousness, generalized weakness, exercise intolerance, fatigue and other symptoms after prolonged standing or sudden postural changes, etc., which can generally be relieved by the supine position [2]. The pathogenesis of POTS is so intricate that it has not been fully understood yet. It is currently known to be associated with autonomic nerve dysfunction, hypovolemia, hyperadrenergic stimulation, deconditioning, hypervigilance and damaged skeletal muscle pump activity, etc., and these mechanisms may exist alone or in combination in patients with POTS [2,3]. In terms of physiological state, the body can appropriately mobilize the nervous and endocrine systems as well as the carotid sinus and the aortic arch baroreceptors to cope with the redistribution of blood volume when the body position changes [3]. POTS patients have excessive blood volume in the lower limbs and visceral beds, which decreases the amount of return blood volume to the heart and cardiac output in the upright position [2,3]. The baroreceptor activity is inhibited and the sympathetic nerve is excited, which increases the heart rate (HR) and cardiac output [3]. However, children with hypovolemia are unable to fully compensate for the decrease in blood volume during postural changes [2,3]. Children in a hyperadrenergic state are characterized by persistent sympathetic excitation and increased HR [2,3]. Moreover, children with POTS tend to have poor exercise and much more anxiety, as well as hypervigilance [2]. POTS is mostly an autonomic nervous dysfunction cardiovascular disease with a relatively good prognosis, but it may cause syncope-related body injury, which not only causes worry and anxiety in children and their families, but also affects the quality of life and learning [3]. Currently, the main diagnostic tool for POTS is the standing test or head-up tilt test (HUTT) [4], but these tests of changing positions can cause convulsions [5], severe arrhythmias [6], temporary aphasia [7] and other complications. It is of clinical significance to explore safe and effective non-invasive test indicators to reflect the diagnosis information of POTS.

Blood pressure variability (BPV) appertains to the degree of blood pressure (BP) fluctuation over a duration of time [8,9]. Standard deviation (SD) and coefficient of variation (CV) are the traditional indices of BPV, which can be measured and obtained by non-invasive 24 h ambulatory blood pressure monitoring (24hABPM) [8,9,10]. BPV generally increases when sympathetic activity increases and during vagus tone withdrawal [8,9]. Sunwoo et al. [11] performed 24hABPM in 103 OI patients (mean age 48.6 ± 17.5 years, 61 females), and divided them into mild and severe groups according to OI questionnaire (OIQ) score. The BPV in the severe OI group (total OIQ score ≥ 12, 47 cases) was higher than in the mild OI group (total OIQ score ≤ 11, 56 cases). Correlation analysis showed that the OI symptoms was positively correlated with 24 h diastolic BPV and daytime diastolic BPV, indicating that BPV can assess the severity of OI patients. The level of BPV in children with POTS has not been reported yet. In this study, we aimed to explore the change in BPV in children with POTS compared with healthy children by 24hABPM. In addition, we attempted to investigate the effect of BPV on POTS and its relationship with prognosis of POTS.

## 2. Materials and Methods

### 2.1. Ethical Statement

This study was approved by the Ethics Committee of The Second Xiangya Hospital, Central South University (Ethical Audit No. Study K034 (2021), 27 October 2021). All parents or guardians of the children signed an informed consent form before participating in the study.

### 2.2. The Studied Subjects

The 47 children with POTS (age 6–14 years, mean age 11.2 ± 1.8 years, 23 males) diagnosed at Pediatric Syncope Clinic or Pediatric Cardiovascular Ward, The Second Xiangya Hospital, Central South University from October 2017 to July 2021 were enrolled as the POTS group, and 30 healthy children (age 6–14 years, mean age 10.9 ± 1.9 years, 15 males) who underwent physical examination at the Children’s Health Specialist Clinic of The Second Xiangya Hospital, Central South University were matched for control group. We excluded 14 children with POTS due to loss of follow-up or incomplete data. According to the intervention effect, 33 patients were divided into response group and non-response group. The POTS group underwent detailed history, physical examination and imaging (chest X-ray, echocardiography, electroencephalogram, head MRI or CT, etc.), blood tests (myocardial enzymes, immune function, fasting plasma glucose, etc.), electrocardiogram (ECG), Holter ECG and other examinations were performed to exclude children with OI signs and symptoms caused by organic heart and brain diseases, metabolic diseases and immune diseases.

### 2.3. HUTT

HUTT is divided into two steps: basic head-up tilt test (BHUT) and sublingual nitroglycerin head-up tilt test (SNHUT). In this study, only BHUT was used to diagnose POTS. HUTT was carried out in the morning [4,12]. The laboratory was equipped with emergency equipment such as oxygen masks and defibrillators. All patients were told to fast and refrain from drinking water for four hours before HUTT, and to not use drugs which might affect the balance of autonomic nervous function for at least five half-lives, and avoid food and drink such as coffee that might influence the autonomic nervous function. During the HUTT, the environment was quiet, the light was dim and the temperature was kept around a comfortable 25 °C. The HUTT was conducted between 8:00 a.m. and 12:00 a.m. Utilizing the Head-up Tilt Test Mornitoring System (SHUT-100) of Jiangsu Standard Medical Technology Co., Ltd., Wuxi City, China, patients were kept supine for 10~30 min on the tilt bed after emptying their bladders, and the BP, HR and ECG were recorded. The tilt bed changed from a supine position to a 60° angle within 15 s, during which the BP, HR, ECG and clinical manifestations of the children were continuously monitored. The HR changes exhibited the most significant magnitude during the initial 10 min of HUTT. In case of syncope or other intolerable symptoms of OI, the test was terminated and the patient was immediately placed in the supine position until recovery.

### 2.4. Diagnosis of POTS

The diagnostic criteria for POTS were as follows [4]: OI occurred during HUTT together with the following responses: (1) The HR increased ≥ 40 bpm during 10 min of HUTT compared to the supine position, or the maximum HR reaches the standard (6~12 years old ≥ 130 bpm, 12~18 years old ≥ 125 bpm), and BP decreased < 20/10 mmHg; (2) Other diseases that cause OI symptoms were excluded.

### 2.5. 24hABPM

The medical technician initialized the ambulatory blood pressure monitoring (ABPM) instrument (ABPM 6100, Welch Allyn, Skaneateles Falls, NY, USA) and input the subject’s personal basic information [13]. The subject’s non-dominant arm (usually the left upper arm) was selected for measurement with an appropriately sized cuff, and the subject was instructed to keep the arm as still as possible during the measurement. BP was manually measured twice at random to confirm that the monitors were in normal working condition. The subjects could go about daily life normally, but were told to avoid anxiety, agitation and strenuous exercise, and ensure that the instruments were still in normal working condition. Subjects were told to record sleep and wake times, OI symptoms such as dizziness and palpitations, activities such as stress and exercise, and taking anti-hypertensive medications that may affect the results of BP measurements. The monitor recorded BP every 30 min during the daytime (7:00~22:00) and nighttime (22:00~7:00). The subjects were instructed to fall asleep at around 22:00 and get up at 7:00 the next day as much as possible. ABPM reports should accord with inclusion criteria [14]. Reports whose overall data did not meet the above criteria were discarded.

### 2.6. Monitoring Indicators

(1) Circadian BP patterns were determined based on the nocturnal dipping of BP: a 10~20% decrease in BP during nighttime was classified as dipping BP, a 0~10% decrease as non-dipping BP, reverse dipping BP if it was less than 0 and extreme dipping BP if it was over 20% [14,15]. (2) According to the reported data, we recorded daytime mean systolic blood pressure (DSBP), daytime systolic blood pressure standard deviation (DSSD), daytime systolic blood pressure variation coefficient (DSCV), daytime mean diastolic blood pressure (DDBP), daytime diastolic blood pressure standard deviation (DDSD), daytime diastolic blood pressure variation coefficient (DDCV), nighttime mean systolic blood pressure (NSBP), nighttime systolic blood pressure standard deviation (NSSD), nighttime systolic blood pressure variation coefficient (NSCV), nighttime mean diastolic blood pressure (NDBP), nighttime diastolic blood pressure standard deviation (NDSD), nighttime diastolic blood pressure variation coefficient (NDCV), 24 h mean systolic blood pressure (24hSBP), 24 h systolic blood pressure standard deviation (24hSSD), 24 h systolic blood pressure variation coefficient (24hSCV), 24 h mean diastolic blood pressure (24hDBP), 24 h diastolic blood pressure standard deviation (24hDSD) and 24 h diastolic blood pressure variation coefficient (24hDCV). Variation coefficient (CV) = SD/mean BP × 100%.

### 2.7. Treatment [4]

(1) Health education: The patients were suggested to avoid predisposing factors such as prolonged standing, sudden changes from lying position, sitting position, or squatting position to upright position. They also should avoid anxiety and take proper exercise. (2) Autonomic nervous function exercise: Patients were encouraged to stand against a wall with feet 15 cm away from the wall, 5 min/episode, twice a day, and gradually increased to 20~30 min/episode. (3) Increase the intake of water and salt: daily water intake of 30~50 mL/kg, oral rehydration salt III (5.125 g every time, dissolved in 250 mL of warm water, twice a day). (4) Metoprolol: The initial dose was 0.5 mg/(kg·d), which was divided into two doses, with a total dose of no more than 2 mg/(kg·d). Oral rehydration salt III (Xi’an Anjian Pharmaceutical Co., Ltd., Xi’an, China, approval number: H20090205, specification: 5.125 g/bag, formula: anhydrous glucose 3.375 g, sodium chloride 0.65 g, sodium citrate 0.725 g, potassium chloride 0.375 g, osmotic pressure 245 mOsm/L, sodium salt concentration 75 mmol/L, tension in half). Metoprolol (Astrazeneca Pharmaceutical (China) Co., Ltd., Shanghai, China, approval number: J20150044, specification: 25 mg/bag).

### 2.8. Prognosis

The 33 patients were followed up for 52.0 (30.5, 90.5) days after receiving four treatments and were divided into two groups as follows: (1) Response to the treatment: the clinical symptoms of the patients were improved (syncope or presyncope or other related symptoms disappeared or lessened), and the reviewed HUTT did not meet the positive criteria at the same time. (2) No-response to the treatment: the clinical symptoms of the patients did not improve significantly (syncope or presyncope or other related symptoms did not disappear or lessen), and the reviewed HUTT met the positive criteria at the same time.

### 2.9. Statistical Analyses

The continuous variables were characterized by mean ± standard deviation. The categorical variables were presented as a number (*n*) and percentages (%). The Student’s *t*-test, Chi-square test, Fisher’s exact test or Mann–Whitney *U* test were conducted to compare factors between different groups, as appropriate. Univariate binary logistic regression was used to briefly evaluate the approximate effects of factors between different groups. Multifactor logistic regression to analyze the possible association between potential influencing factors and outcomes was used, and different models were constructed to illustrate the stability of this relationship. All the analyses were performed with the statistical software packages R (version 4.1) (http://www.R-project.org, The R Foundation, accessed on 10 July 2023) and EmpowerStats (http://www.empowerstats.com (accessed on 10 July 2023), X & Y Solutions, Inc, Boston, MA, USA). *p*-values < 0.05 (two-sided) were considered statistically significant.

## 3. Results

### 3.1. Comparison of General Data and Ambulate Blood Pressure and BPV between the POTS Group and Control Group

There were no significant differences in demographic factors and mean BP and BP pattern between the POTS group and the control group (*p* > 0.05). The 24hDSD, DDSD, NSSD, DDCV and NSCV in the control group were lower than in the POTS group (*p* < 0.05), as shown in Table 1.

### 3.2. Univariate Binary Logistic Regression Analysis for POTS

After univariate analysis of all variables, we found that 24hDSD, DDSD, NSSD, DDCV and NSCV were potential risk factors for children with POTS. For every 1 mmHg increase in 24hDSD, DDSD and NSSD, the risk of POTS increased by 55%, 51% and 47%, respectively. For every 1% increase in DDCV and NSCV, the risk of POTS increased by 21% and 44%, respectively, as shown in Table 2.

### 3.3. Comparison of Multiple Multifactor Binary Logistic Regression for POTS

To find out whether the potential risk variables obtained by univariate analysis were independent, we adjusted the number of covariates in the argument. After adjusting for sociodemographic factors, 24hDSD and DDCV were no longer significant risk factors for POTS, indicating that they were not independent risk factors for POTS. The potential risk effects of DDSD, NSSD and NSCV on POTS were still significant, indicating that they were independent risk factors for POTS. The risk of POTS increased by 48%, 53% and 49% for every 1 mmHg, 1 mmHg and 1% increase in DDSD, NSSD and NSCV, respectively, as shown in Table 3.

### 3.4. Comparison of General Data and Ambulate Blood Pressure and BPV between Response Group and Non-Response Group in Children with POTS

Percentage of females, age and height in the POTS response group were lower than in the non-response group (*p* < 0.05). There were no significant differences in mean BP, baseline BPV and other factors between the POTS response group and no-response group (*p* > 0.05), as shown in Table 4.

### 3.5. Univariate Binary Logistic Regression Analysis for Poor Prognosis in Children with POTS

After univariate analysis of all variables, we found that the risk of poor prognosis in females was 4.71 times higher than that in males. For every 1 cm increase in height, the risk of poor prognosis in POTS increased by 7%. Sex and height might be potential risk factors for poor prognosis in children with POTS, as shown in Table 5.

### 3.6. Comparison of Multiple Multifactor Binary Logistic Regression for Poor Prognosis in Children with POTS

To explore whether the potential risk variables obtained by univariate analysis were independent, we adjusted for sociodemographic and other confounding factors. Sex, but not height, was still a potential risk for poor prognosis of POTS. The risk of poor prognosis of POTS in females was 12.99 times higher than that in males, indicating that sex was an independent risk factor for poor prognosis of POTS, as shown in Table 6.

## 4. Discussion

In terms of physiological conditions, BP has circadian rhythms to protect cardiovascular structure and function, and a dipping BP represents a normal circadian rhythm [14]. Non-dipping and reverse dipping BP circadian rhythms are related to the risk of cardiovascular outcome and target organ damage [15]. The patients with non-dipping BP pattern showed a non-significant nocturnal decline in epinephrine (E) and norepinephrine (NE) excretion, and the sympathetic activity showed insufficient decline in the nighttime [16]. The main mechanism of POTS is autonomic nerve dysfunction due to increased sympathetic excitation resulting from elevated plasma NE levels [17]. Cai et al. [18] performed 24hABPM in 103 children with POTS and 84 healthy children, and found that the non-dipping BP pattern in POTS patients was more frequent than healthy children (67% vs. 46%, *p* < 0.01). In our study, the POTS group showed mainly a non-dipping BP pattern compared with the control group (70.21% vs. 56.67%, *p* = 0.224), suggesting that autonomic nerve function might be disordered and that the sympathetic nerve was relatively excited during the night. No statistically significant difference was seen as related to the small sample size. The 24hABPM can dynamically monitor BP according to the set interval, can be used for comprehensive analysis of daytime and nighttime BP, and can distinguish between white coat hypertension, masked hypertension, prehypertension, etc. It is simple and non-invasive [13,14].

BPV refers to the fluctuation level of BP within a period of time and is a non-invasive index used to evaluate autonomic nervous function, which can be divided into short-term, medium-term, long-term and seasonal BPV, according to different monitoring times [9]. Our POTS group aimed at short-term BPV studies based on objective conditions. A decrease in BP at night is correlated with a marked drop in sympathetic drives, and an increase in BP during the day is linked to sympathetic activation [9]. The increase in BPV may reflect impaired baroreflex function and enhanced sympathetic drive [9]. The study showed that 24hDSD, DDSD, NSSD, DDCV and NSCV in the control group were lower than in the POTS group (*p* < 0.05). DDSD, NSSD and NSCV were independent risk factors for POTS. Compared with healthy children, children with POTS had higher sympathetic excitability throughout the day, which might be the result of a combination of complex pathophysiological mechanisms of POTS. Our recent study also showed that NSSD, NDSD and NSCV in the control group were lower than in the vasoinhibitory-type vasovagal syncope (VVS-VI) group [19], indicating increased sympathetic activity at night in children with VVS-VI. Onishi et al. [20] performed 24hABPM in 40 HUTT-positive patients (mean age 46.2 ± 22.9 years, 17 males) and 48 HUTT-negative patients (mean age 46.8 ± 18.7 years, 34 males), and found the NSSD of HUTT-positive group and HUTT-negative group, along with the NDSD of the HUTT-negative group were higher than that of the control group (12.4 ± 4.1 mmHg vs. 11.8 ± 3.6 mmHg vs. 9.56 ± 3.5 mmHg, 9.99 ± 2.94 mmHg vs. 7.82 ± 2.64 mmHg, *p* < 0.05), indicating that the autonomic nervous dysfunction existed in the neurally mediated syncope (NMS) patients and the nighttime sympathetic tone increased regardless of the HUTT results.

Signs of POTS patients were mainly characterized by a significant increase in HR in the upright position and a small change in BP, which might account for the statistically significant but small difference in BPV between the POTS and control groups. Our previous study found that NDSD and NDCV could predict the prognosis of VVS-VI [19]. This study showed that BPV showed no statistical difference between the response group and the non-response group in children with POTS. The main sign of VVS-VI was a decrease in BP without significant change in HR. The results might be related to the difference in the signs of POTS and VVS-VI. However, the influence of other factors cannot be excluded. POTS patients often present with the coexistence of multiple mechanisms [2,3]. Peripheral pooling of blood and hypovolemia may be related to peripheral denervation, blood accumulation in the lower limbs and viscera beds, or abnormal activation and secretion of the renin–angiotensin system [2,3,21]. Half of all POTS patients show a hyperadrenergic state with NE ≥ 600 pg/mL in the upright position [2,18,21]. POTS patients often have poor exercise tolerance and deconditioning, and have higher anxiety and somatic vigilance [2]. In addition to increased nocturnal sympathetic activity, factors such as reduced renal sodium excretion, leptin and insulin resistance, salt sensitivity, endothelial dysfunction and sleep breathing disorders affect the non-dipping and reverse dipping BP pattern [9]. Previous studies have shown that age, mean BP, BMI, waist circumference, poor sleep quality, exercise, stress and smoking had a positive correlation with BPV, and sex and ethnic difference may affect BPV [22,23,24,25]. Some studies also showed that BMI, age, sex, birth weight, race, neuroendocrine axis, medication, sleep quality and so on were related to BP levels, which can possibly affect the BPV [3,14]. The above factors and other unknown factors may affect the acquisition and comparative judgment of BPV. The study showed that the risk of poor prognosis of POTS in female was 12.99 times higher than that in male, indicating that sex was an independent risk factor for poor prognosis of POTS. There were studies showing that the higher incidence of OI in females was due to cardiac mechanics (females have a smaller left ventricular cavity than males, possibly due to higher systolic elasticity and lower diastolic compliance). This may be due to the Frank–Starling relationship (the maximum slope of the Starling curve is steeper in females than in males. Stroke volume, stroke index and cardiac filling in females are lower than those in males under normovolemic or hypovolemic conditions at presyncope), sexual dimorphism of muscle sympathetic nerve discharge (females are more active during hypotensive stimulation), and differences in hormone secretion (female estrogen plays an important role in vagus nerve regulation) [26,27,28,29]. Whether those factors lead to poor prognosis in female patients needs further study.

BPV is relevant to an increased prevalence and progression of cardiac, vascular, renal and other organ damage [9]. Current studies have found that some drug treatments and non-drug treatments may reduce BP and BPV. Jiang et al. [30] used the angiotensin type I receptor blocker losartan and the calcium channel blocker azelnidipine to treat experimental rats with an increase in mean BP and BPV caused by subcutaneous infusion of angiotensin II. This can reduce the mean BP and BPV, suggesting that an abnormal renin–angiotensin–aldosterone system can lead to an increase in BPV, and angiotensin II increases BPV in an angiotensin-type-I-receptor-dependent manner. Mitro et al. [31] found that baroreflex sensitivity (BRS) increased after orthostatic training in patients with vasovagal syncope, and the upright BRS of non-responders was lower than that of responders, and the relationship between BRS and BPV was negative [32], so it was possible to reduce BPV. Treatments for POTS such as health education, orthostatic training, increasing blood volume by salt and water intake and metoprolol may reduce BPV and thus reduce long-term risk. It is a pity that no analysis was conducted of ambulatory blood pressure before and after treatment to verify the change in BPV due to objective reasons such as the small number of patients who were followed up, the smaller number of patients who reviewed ambulatory blood pressure after treatment, and the incomplete data.

In this study, BPV might have increased in children with POTS compared with healthy children. DDSD, NSSD and NSCV are independent risk factors for POTS, and female sex is an independent risk factor for poor prognosis of POTS. On the one hand, although we avoided correlated interference as much as possible during the experimental design phase, the study had limitations. We need to increase the sample size, optimize the experimental design to further verify the level of BPV in children with POTS. We should also follow up the children after treatment and review indicators to explore the changes of BPV. On the other hand, BPV can be obtained by non-invasive ABPM, which may become a new auxiliary indicator to assist the diagnosis and evaluate the therapeutic effect of OI.

## 5. Limitations

The study had the following limitations: (1) 24hABPM was performed only once while the ambulatory BP is affected by many factors, which might influence the accuracy of the measurement data. (2) POTS patients often express sleepiness, fatigue, frequent arousal from sleep, but sleep quality was not analyzed in the study, which can affect BPV. (3) The sample size was small, which might affect the final judgement. (4) Changes in BPV after treatment and their comparison with healthy children were not analyzed.

## 6. Conclusions

Although our study has some limitations, it suggests that children with POTS may have an abnormal blood pressure circadian rhythm and increased BPV compared to that of healthy children. DDSD, NSSD and NSCV are independent risk factors for POTS, and female sex is an independent risk factor for poor prognosis of POTS. Changes of BPV in children with POTS need to be further studied.

## Figures and Tables

**Table 1 children-10-01244-t001:** Comparison of general data and ambulate blood pressure and BPV between the POTS group and the control group (Mean ± SD, *n* (%)).

Characteristics	Control Group	POTS Group	Standardize Diff	*p*-Value
Comparison of demographic data and BP pattern				
*N*	30	47		
Sex			0.02 (−0.44, 0.48)	0.927
Male	15 (50.00)	23 (48.94)		
Female	15 (50.00)	24 (51.06)		
Age, years	10.93 ± 1.89	11.21 ± 1.83	0.15 (−0.31, 0.61)	0.521
Height, cm	147.50 ± 12.28	152.56 ± 12.94	0.40 (−0.06, 0.86)	0.092
Weight, kg	37.72 ± 10.84	39.50 ± 9.42	0.18 (−0.28, 0.63)	0.447
BMI, kg/m^2^	16.98 ± 2.63	16.72 ± 2.11	0.11 (−0.35, 0.57)	0.632
BP pattern			0.28 (−0.18, 0.74)	0.224
Dipping	13 (43.33)	14 (29.79)		
Non-dipping	17 (56.67)	33 (70.21)		
Comparison of the baseline ambulate blood pressure				
24hSBP, mmHg	101.33 ± 6.38	102.57 ± 5.75	0.20 (−0.25, 0.66)	0.379
24hDBP, mmHg	55.53 ± 3.84	56.62 ± 4.08	0.27 (−0.19, 0.73)	0.249
DSBP, mmHg	104.53 ± 6.62	105.55 ± 6.04	0.16 (−0.30, 0.62)	0.489
DDBP, mmHg	58.77 ± 4.21	59.72 ± 4.51	0.22 (−0.24, 0.68)	0.355
NSBP, mmHg	95.60 ± 7.02	97.06 ± 6.57	0.22 (−0.24, 0.67)	0.356
NDBP, mmHg	49.90 ± 4.21	50.66 ± 4.06	0.18 (−0.28, 0.64)	0.432
Comparison of the baseline BPV				
24hSSD, mmHg	8.68 ± 1.64	9.11 ± 1.43	0.28 (−0.18, 0.75)	0.219
24hDSD, mmHg	8.19 ± 1.03	8.97 ± 1.56	0.58 (0.12, 1.05)	0.019
DSSD, mmHg	7.83 ± 1.64	8.26 ± 1.60	0.26 (−0.20, 0.72)	0.267
DDSD, mmHg	7.41 ± 1.31	8.39 ± 1.84	0.62 (0.15, 1.08)	0.013
NSSD, mmHg	6.32 ± 1.70	7.40 ± 1.84	0.61 (0.14, 1.08)	0.011
NDSD, mmHg	5.69 ± 1.35	6.32 ± 1.34	0.47 (0.00, 0.93)	0.049
24hSCV, %	8.58 ± 1.64	8.90 ± 1.39	0.21 (−0.25, 0.67)	0.362
24hDCV, %	14.79 ± 1.88	15.89 ± 2.84	0.46 (−0.01, 0.92)	0.065
DSCV, %	7.50 ± 1.54	7.84 ± 1.53	0.22 (−0.24, 0.68)	0.352
DDCV, %	12.66 ± 2.37	14.11 ± 3.14	0.52 (0.06, 0.99)	0.033
NSCV, %	6.61 ± 1.72	7.62 ± 1.77	0.58 (0.11, 1.05)	0.015
NDCV, %	11.42 ± 2.62	12.55 ± 2.78	0.42 (−0.04, 0.88)	0.080

**Table 2 children-10-01244-t002:** Univariate binary logistic regression analysis for POTS (Mean ± SD, *n* (%)).

	Statistics	OR (95% CI)	*p*-Value
Sex			
Male	38 (49.35)	1.0	
Female	39 (50.65)	1.04 (0.42, 2.61)	0.928
Age, years	11.10 ± 1.85	1.09 (0.85, 1.39)	0.516
Height, cm	150.59 ± 12.85	1.03 (0.99, 1.07)	0.095
Weight, kg	38.81 ± 9.97	1.02 (0.97, 1.07)	0.442
BMI, kg/m^2^	16.82 ± 2.31	0.95 (0.78, 1.16)	0.627
BP pattern			
Dipping	27 (35.06)	1.0	
Non-dipping	50 (64.94)	1.80 (0.69, 4.68)	0.227
24hSBP, mmHg	102.09 ± 5.99	1.04 (0.96, 1.12)	0.375
24hDBP, mmHg	56.19 ± 4.00	1.07 (0.95, 1.21)	0.247
DSBP, mmHg	105.16 ± 6.25	1.03 (0.95, 1.11)	0.484
DDBP, mmHg	59.35 ± 4.39	1.05 (0.95, 1.17)	0.350
NSBP, mmHg	96.49 ± 6.74	1.03 (0.96, 1.11)	0.352
NDBP, mmHg	50.36 ± 4.11	1.05 (0.93, 1.17)	0.428
24hSSD, mmHg	8.94 ± 1.52	1.22 (0.89, 1.67)	0.218
24hDSD, mmHg	8.66 ± 1.42	1.55 (1.06, 2.27)	0.024
DSSD, mmHg	8.09 ± 1.62	1.18 (0.88, 1.59)	0.265
DDSD, mmHg	8.01 ± 1.71	1.51 (1.07, 2.12)	0.018
NSSD, mmHg	6.98 ± 1.85	1.47 (1.07, 2.01)	0.016
NDSD, mmHg	6.08 ± 1.37	1.45 (0.99, 2.12)	0.055
24hSCV, %	8.78 ± 1.49	1.16 (0.85, 1.59)	0.357
24hDCV, %	15.46 ± 2.55	1.21 (0.98, 1.48)	0.071
DSCV, %	7.70 ± 1.53	1.16 (0.85, 1.58)	0.348
DDCV, %	13.55 ± 2.94	1.21 (1.01, 1.45)	0.038
NSCV, %	7.23 ± 1.81	1.44 (1.06, 1.95)	0.020
NDCV, %	12.11 ± 2.75	1.17 (0.98, 1.41)	0.084

Result variable: POTS. Exposure variable: sex, age, height, weight, BMI, BP pattern, 24hSBP, 24hDBP, DSBP, DDBP, NSBP, NDBP, 24hSSD, 24hDSD, DSSD, DDSD, NSSD, NDSD, 24hSCV, 24hDCV, DSCV, DDCV, NSCV, NDCV. Adjusted for: None.

**Table 3 children-10-01244-t003:** Comparison of multiple multifactor binary logistic regression for POTS.

Exposure	OR (95% CI)	*p*-Value
24hDSD	1.61 (0.99, 2.61)	0.056
DDSD	1.48 (1.01, 2.17)	0.045
NSSD	1.53 (1.04, 2.25)	0.031
DDCV	1.21 (0.99, 1.49)	0.068
NSCV	1.49 (1.02, 2.18)	0.038

Result variable: POTS. Exposure variable: 24hDSD, DDSD, NSSD, DDCV, NSCV. Model adjusted for: sex, age, height, weight.

**Table 4 children-10-01244-t004:** Comparison of general data and ambulate blood pressure and BPV between response group and non-response group in children with POTS (Mean ± SD, *n* (%)).

Characteristics	Response Group	Non-Response Group	Standardize Diff	*p*-Value
Comparison of demographic data and BP pattern				
*N*	18	15		
Sex			0.83 (0.12, 1.55)	0.026
Male	15 (83.33)	7 (46.67)		
Female	3 (16.67)	8 (53.33)		
Age, years	10.72 ± 2.24	12.13 ± 1.36	0.76 (0.05, 1.47)	0.041
Height, cm	149.56 ± 14.96	159.70 ± 8.96	0.82 (0.11, 1.54)	0.028
Weight, kg	38.67 ± 11.11	43.50 ± 7.66	0.51 (−0.19, 1.20)	0.164
BMI, kg/m^2^	16.89 ± 2.26	16.95 ± 2.01	0.03 (−0.66, 0.71)	0.938
BP pattern			0.40 (−0.29, 1.09)	0.261
Dipping	5 (27.78)	7 (46.67)		
Non-dipping	13 (72.22)	8 (53.33)		
Comparison of the baseline ambulate blood pressure				
24hSBP, mmHg	102.67 ± 5.63	103.53 ± 7.24	0.13 (−0.55, 0.82)	0.701
24hDBP, mmHg	56.56 ± 4.96	57.53 ± 3.98	0.22 (−0.47, 0.90)	0.543
DSBP, mmHg	105.78 ± 5.39	106.80 ± 7.94	0.15 (−0.54, 0.84)	0.664
DDBP, mmHg	59.83 ± 5.19	60.93 ± 4.62	0.22 (−0.46, 0.91)	0.529
NSBP, mmHg	97.06 ± 6.55	97.27 ± 8.25	0.03 (−0.66, 0.71)	0.935
NDBP, mmHg	50.61 ± 4.42	50.40 ± 4.82	0.05 (−0.64, 0.73)	0.897
Comparison of the baseline BPV				
24hSSD, mmHg	9.16 ± 1.18	9.24 ± 1.76	0.05 (−0.63, 0.74)	0.879
24hDSD, mmHg	9.28 ± 1.44	9.31 ± 1.96	0.02 (−0.66, 0.71)	0.953
DSSD, mmHg	8.44 ± 1.70	7.98 ± 1.49	0.29 (−0.40, 0.98)	0.415
DDSD, mmHg	8.86 ± 2.00	8.57 ± 2.08	0.14 (−0.54, 0.83)	0.682
NSSD, mmHg	7.07 ± 1.37	7.41 ± 1.79	0.21 (−0.47, 0.90)	0.540
NDSD, mmHg	6.29 ± 1.17	5.87 ± 1.13	0.36 (−0.33, 1.05)	0.311
24hSCV, %	8.94 ± 1.18	8.95 ± 1.68	0.00 (−0.68, 0.69)	0.992
24hDCV, %	16.47 ± 2.63	16.28 ± 3.69	0.06 (−0.63, 0.75)	0.862
DSCV, %	7.99 ± 1.57	7.50 ± 1.45	0.32 (−0.37, 1.01)	0.365
DDCV, %	14.88 ± 3.39	14.14 ± 3.61	0.21 (−0.48, 0.90)	0.551
NSCV, %	7.29 ± 1.31	7.63 ± 1.76	0.21 (−0.47, 0.90)	0.540
NDCV, %	12.53 ± 2.66	11.82 ± 2.88	0.26 (−0.43, 0.94)	0.469

**Table 5 children-10-01244-t005:** Univariate binary logistic regression analysis for poor prognosis in children with POTS (Mean ± SD, *n* (%)).

	Statistics	OR (95% CI)	*p*-Value
Sex			
Male	22 (66.67)	1.0	
Female	11 (33.33)	5.71 (1.15, 28.35)	0.033
Age, years	11.36 ± 2.00	1.60 (0.98, 2.60)	0.060
Height, cm	154.17 ± 13.43	1.07 (1.00, 1.15)	0.041
Weight, kg	40.86 ± 9.86	1.06 (0.98, 1.14)	0.166
BMI, kg/m^2^	16.92 ± 2.12	1.01 (0.73, 1.41)	0.936
BP pattern			
Dipping	12 (36.36)	1.0	
Non-dipping	21 (63.64)	0.44 (0.10, 1.87)	0.265
24hSBP, mmHg	103.06 ± 6.32	1.02 (0.92, 1.14)	0.691
24hDBP, mmHg	57.00 ± 4.50	1.05 (0.90, 1.23)	0.530
DSBP, mmHg	106.24 ± 6.58	1.02 (0.92, 1.14)	0.653
DDBP, mmHg	60.33 ± 4.90	1.05 (0.91, 1.21)	0.516
NSBP, mmHg	97.15 ± 7.25	1.00 (0.91, 1.11)	0.933
NDBP, mmHg	50.52 ± 4.54	0.99 (0.85, 1.15)	0.893
24hSSD, mmHg	9.20 ± 1.45	1.04 (0.64, 1.68)	0.874
24hDSD, mmHg	9.29 ± 1.67	1.01 (0.67, 1.54)	0.951
DSSD, mmHg	8.23 ± 1.60	0.83 (0.53, 1.29)	0.404
DDSD, mmHg	8.73 ± 2.01	0.93 (0.65, 1.32)	0.671
NSSD, mmHg	7.23 ± 1.56	1.16 (0.74, 1.82)	0.528
NDSD, mmHg	6.10 ± 1.16	0.71 (0.37, 1.36)	0.306
24hSCV, %	8.94 ± 1.41	1.00 (0.61, 1.64)	0.992
24hDCV, %	16.38 ± 3.11	0.98 (0.78, 1.23)	0.857
DSCV, %	7.77 ± 1.51	0.80 (0.50, 1.28)	0.354
DDCV, %	14.54 ± 3.46	0.94 (0.76, 1.15)	0.539
NSCV, %	7.45 ± 1.51	1.16 (0.73, 1.85)	0.528
NDCV, %	12.21 ± 2.74	0.91 (0.70, 1.18)	0.457

Result variable: poor prognosis. Exposure variable: sex, age, height, weight, BMI, 24hSBP, 24hDBP, DSBP, DDBP, NSBP, NDBP, 24hSSD, 24hDSD, DSSD, DDSD, NSSD, NDSD, 24hSCV, 24hDCV, DSCV, DDCV, NSCV, NDCV. Adjusted for: None.

**Table 6 children-10-01244-t006:** Comparison of multifactor binary logistic regression models for poor prognosis in children with POTS.

	Sex			Height	
	OR (95% CI)	*p*-Value		OR (95% CI)	*p*-Value
Model 1	1.0			1.14 (0.96, 1.36)	0.125
	9.88 (1.28, 76.46)	0.028			
Model 2	1.0			1.18 (0.92, 1.52)	0.197
	13.99 (1.13, 173.39)	0.040			

Result variable: poor prognosis. Exposure variables: sex, height. Model 1 of sex adjusted for age, height, weight. Model 2 of sex adjusted for age, height, weight, 24hSSD, 24hDSD, 24hSCV, 24hDCV, 24hSBP, 24hDBP. Model 1 of height adjusted for age, sex, weight. Model 2 of height adjusted for age, sex, weight, 24hSSD, 24hDSD, 24hSCV, 24hDCV, 24hSBP, 24hDBP.

## Data Availability

The data presented in this study are available on request from the corresponding author: Cheng Wang.

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
