# Peer review of "Changes in Blood Pressure Variability in Children with Postural Tachycardia Syndrome"

_children, 2023, doi:10.3390/children10071244_

Round 1

Reviewer 1 Report

The authors objectively compared blood pressure variability and other parameters in children diagnosed with POTS with a healthy control group. The results of this study seem valuable in terms of shedding light on the pathophysiology of POTS. The authors presented the manuscript well. I think that the study will contribute to the literature. It would be more appropriate to present the weight, height and mean blood pressure (systolic and diastolic) values ​​in the table for POTS and control groups.

Author Response

Thanks for the suggestion, we have made tables 1 and 2 to show the comparison of general data and mean blood pressure between the study group and the control group.

Reviewer 2 Report

1) Citations 15 is incorrectly interpreted in the manuscript ( lines 148-150). In non dippers there is insufficient decline in NE and E levels during the night (not significant decline as stated in the manuscript). 

2) Please rephrase sentense in the line 200 and 201: (To the extent .....) . The sentence meaning is not clear. 

3) Not all parameters of BPV are significantly different in POTS patients compared to controls . Thus sattement about  increased BPV in these subjects should be more caution  - they may have increased BPV. 

Author Response

Question1:Citations 15 is incorrectly interpreted in the manuscript (lines 148-150). In non-dippers there is insufficient decline in NE and E levels during the night (not significant decline as stated in the manuscript).

Answer: Thanks for the suggestion, I have corrected my mistakes.

Question 2: Please rephrase sentence in the line 200 and 201(To the extent…).The sentence meaning is not clear.

Answer: Thanks for the suggestion, I have modified this sentence to“The treatment for POTS such as health education, orthostatic training, increasing blood volume by salt and water intake, and metoprolol may reduce BPV and thus reduce long-term risk.”

Question 3: Not all parameters of BPV are significantly different in POTS patients compared to controls. Thus statement about increased BPV in these subjects should be more caution-they may have increased BPV.

Answer: Thanks for the suggestion, we have modified the statement more cautiously.

Reviewer 3 Report

This article by Gu et al studies the circadian blood pressure rhythm in POTS patients.

This is an interesting and well written article. Some minor concerns are listed below.

The study group is very small (47 patients) and this is an important limitation.

Discussions section: “The POTS group was mainly a non-dipping BP pattern compared with the control group, suggesting that autonomic nerve function is disordered and sympathetic nerve was relatively excited during the night”- please provide the exact data, as “mainly” is a vague term.

Author Response

Question1:This article by Gu et al studies the circadian blood pressure rhythm in POTS patients. This is an interesting and well written article. Some minor concerns are listed below.

The study group is very small (47 patients) and this is an important limitation.

Answer: Thanks for the suggestion, Due to the low compliance rate of ambulatory blood pressure, the final included sample size was small. I will try to optimize the experimental design and improve the coincidence rate in the future research.

Question2: Discussions section:“The POTS group was mainly a non-dipping BP pattern compared with the control group, suggesting that autonomic nerve function is disordered and sympathetic nerve was relatively excited during the night”-please provide the exact data. As“mainly”is a vague term.

Answer: Thanks for the suggestion, we have amended this sentence and added exact data.
